# Neurosteroids Mediate Neuroprotection in an In Vitro Model of Hypoxic/Hypoglycaemic Excitotoxicity via δ-GABA_A_ Receptors without Affecting Synaptic Plasticity

**DOI:** 10.3390/ijms24109056

**Published:** 2023-05-21

**Authors:** Xènia Puig-Bosch, Markus Ballmann, Stefan Bieletzki, Bernd Antkowiak, Uwe Rudolph, Hanns Ulrich Zeilhofer, Gerhard Rammes

**Affiliations:** 1Department of Anaesthesiology and Intensive Care Medicine, Medical School, Klinikum Rechts der Isar, Technical University Munich, 81675 Munich, Germany; markus.ballmann@tum.de (M.B.); g.rammes@tum.de (G.R.); 2BCN-AIM Lab, Departament de Matemàtiques i Informàtica, Universitat de Barcelona, 08007 Barcelona, Spain; 3Department of Anaesthesiology and Intensive Care Medicine, Experimental Anaesthesiology Section, Eberhard Karls University, 72072 Tübingen, Germany; stefan.bieletzki@uni-tuebingen.de (S.B.); bernd.antkowiak@uni-tuebingen.de (B.A.); 4Department of Comparative Biosciences, and Carl R. Woese Institute for Genomic Biology, College of Veterinary Medicine, University of Illinois at Urbana-Champaign, Urbana, IL 61801, USA; urudolph@illinois.edu; 5Institute of Pharmacology and Toxicology, University of Zürich, 8057 Zürich, Switzerland; zeilhofer@pharma.uzh.ch; 6Institute of Pharmaceutical Sciences, Swiss Federal Institute of Technology (ETH) Zürich, 8092 Zürich, Switzerland

**Keywords:** excitotoxicity, GABA_A_ receptors, hippocampus, LTP, neuroprotection, neurosteroids, TSPO, XBD173

## Abstract

Neurosteroids and benzodiazepines are modulators of the GABA_A_ receptors, thereby causing anxiolysis. Furthermore, benzodiazepines such as midazolam are known to cause adverse side-effects on cognition upon administration. We previously found that midazolam at nanomolar concentrations (10 nM) blocked long-term potentiation (LTP). Here, we aim to study the effect of neurosteroids and their synthesis using XBD173, which is a synthetic compound that promotes neurosteroidogenesis by binding to the translocator protein 18 kDa (TSPO), since they might provide anxiolytic activity with a favourable side-effect profile. By means of electrophysiological measurements and the use of mice with targeted genetic mutations, we revealed that XBD173, a selective ligand of the translocator protein 18 kDa (TSPO), induced neurosteroidogenesis. In addition, the exogenous application of potentially synthesised neurosteroids (THDOC and allopregnanolone) did not depress hippocampal CA1-LTP, the cellular correlate of learning and memory. This phenomenon was observed at the same concentrations that neurosteroids conferred neuroprotection in a model of ischaemia-induced hippocampal excitotoxicity. In conclusion, our results indicate that TSPO ligands are promising candidates for post-ischaemic recovery exerting neuroprotection, in contrast to midazolam, without detrimental effects on synaptic plasticity.

## 1. Introduction

Benzodiazepines are fast-acting anxiolytics and widely used in premedication [1], anaesthesia induction [2], sedation in the intensive care unit [3], and in procedural anaesthesia outside of the operational theatre [4]. However, benzodiazepines also produce deleterious effects on memory [5] and may also induce postoperative neurocognitive disorder [6], specifically affecting the senescent and developing brain [7,8].

Stroke, epilepsy, and traumatic brain injury share complex neuropathological mechanisms that produce acute and chronic disturbances of brain function. One of the sequelae is the development of cognitive impairment, which may develop into long-term dysfunction [9]. Therefore, treatment with benzodiazepines bears the risk of exaggerating cognitive dysfunction [10,11].

Midazolam at low doses is used as an anxiolytic premedication in human subjects. Unfortunately, midazolam at anxiolytic doses blocks hippocampal long-term potentiation (LTP) [12], a cellular correlate for learning and memory, imparted mainly via α1-GABA_A_ receptors [13]. This implies, combined with its pharmacokinetic profile, that even more than 12 h after terminating midazolam treatment, midazolam may produce cognitive dysfunction [6,14]. Therefore, there is an unmet need for fast-acting anxiolytic drugs with less severe effects on memory.

3α-reduced neurosteroids such as allopregnanolone and THDOC, at relevant and low physiological concentrations (10–100 nM), are potent positive allosteric modulators of GABA_A_ receptors [15,16], presenting their highest affinity for δ-containing receptors [17,18]. Moreover, evidence shows that the high-affinity translocator protein 18 kDa (TSPO) ligand XBD173 induces neurosteroidogenesis and exerts rapid anxiolytic effects in animal models and humans in the absence of sedation, tolerance development, and withdrawal symptoms [19,20]. Furthermore, neurosteroids and TSPO ligands have been shown to exert neuroprotection in models of ischaemia [21,22].

Here, we aim to elucidate the neuroprotective efficacy of the specific TSPO ligand XBD173 in comparison to the benzodiazepine midazolam. Since benzodiazepines and neurosteroids target different molecular binding sites, we intend to identify the GABA_A_-receptor subunits involved in XBD173 action. Furthermore, we are interested in the action of the neurosteroids putatively synthesised upon XBD173-induced TSPO activation.

Stroke, epilepsy, and traumatic brain injury are accompanied by perturbations of energy homeostasis and a significant increase in extracellular glutamate [23]. In this study, we used a model of excitotoxicity (hypoxia/hypoglycaemia; H/H) which can probably be linked to the in vivo neuropathology of these diseases when glucose and oxygen supplies are reduced or even eliminated.

## 2. Results

### 2.1. Neuroprotective Properties against Ischaemic-Induced Excitotoxicity

We examined whether XBD173-induced TSPO activation conferred protection to hippocampal brain slices subjected to H/H. XBD173 (300 nM) presented for 60 min before the ischaemic period and continuously thereafter significantly blocked the suppression of the field excitatory postsynaptic potential (fEPSP) slope (112(82) % vs. control 54(15) %, *n* = 6/6, *p* = 0.002) in wild-type (WT) animals (Figure 1A,B). In contrast, when XBD173 at 300 nM was administered in δ-knock-out (KO) (24(16) % vs. control 21(19) %, *n* = 6/6, *p* = 0.132) and in TSPO-KO slice genotypes (70(35) % vs. control 51(23) %, *n* = 6/6, *p* = 0.898) (Figure 1C,D, respectively), fEPSP slope did not recover. Thus, we were able to show that XBD173 is neuroprotective in WT mice, but this effect is lost in δ-KO and TSPO-KO mice. 

TSPO activation by XBD173 promotes the synthesis of endogenous neurosteroids such as THDOC and allopregnanolone [20,24]. To determine whether the neurosteroid THDOC may antagonise H/H-induced excitotoxicity, we firstly tested this neurosteroid in WT animals (Figure 2A). THDOC at 100 nM significantly reversed the effect of ischaemic-induced excitotoxicity (103(20) % vs. control 60(16) %, *n* = 6/6, *p* = 0.004) (Figure 2B) since the fEPSP slope and responses recovered to baseline levels. This recovery was completely absent in δ-KO slices and fEPSP remained reduced after washing in THDOC at 100 nM (Figure 2C), similar to control values (21(16) % vs. control 19(11) %, *n* = 8/8, *p* = 0.375). Secondly, allopregnanolone administered to slices from WT mice at 100 nM resulted in a significant fEPSP slope recovery (111(15) % vs. control 42(22) %, *n* = 6/6, *p* = 0.002) after its exposure in the ischaemia model (Figure 2D,E), thereby causing neuroprotection. Nevertheless, this recovery was absent when administering allopregnanolone in δ-KO mice (48(40) % vs. control 55(49) %, *n* = 6/6, *p* > 0.999) (Figure 2F). Hence, similarly to THDOC, allopregnanolone triggers neuroprotection in WT mice, yet loses its effect when the δ subunit is missing in the GABA_A_ receptor. 

Moreover, at the low nanomolar concentration of 10 nM, the benzodiazepine midazolam significantly blocked excitotoxicity induced by H/H in WT animals (100(29) % vs. control 59(15) %, *n* = 8/7, *p* = 0.001) (Figure 3A,B), as well as in slices from α1/2/3-knock-in (KI) mice (95(44) % vs. control 59(53) %, *n* = 8/8, *p* = 0.028) (Figure 3C), a genotype in which only α5 subtype is sensitive to modulation by midazolam. Midazolam has not been tested in the δ-KO model because δ-containing GABA_A_ receptors lacking a γ-subunit are insensitive to benzodiazepines. Thus, we opted for the α1/2/3-KI line because α5 receptors are similar to δ receptors located extrasynaptically in the hippocampus. Nevertheless, when slices from α5-KI mice were perfused with midazolam at 10 nM (only α5 receptors are insensitive to the drug), after ischaemia, the fEPSP slope did not recover to baseline levels (53(5) % vs. control 46(7) %, *n* = 8/7, *p* = 0.001) although a significant increase was apparent (Figure 3D).

### 2.2. Effects of XBD173, THDOC and Allopregnanolone on LTP

LTP is the principal ex vivo model used to investigate synaptic plasticity in hippocampal brain slices since it is considered to be the cellular correlate for learning and memory processes [25] and it serves as an ex vivo model of a physiological neuronal process. Additionally, its inhibition is associated with cognitive impairment [26]. Here, we aimed to clarify whether the concentrations of the TSPO ligand XBD173 and neurosteroids that conferred neuroprotection in the oxygen–glucose deprivation model cause detrimental effects after LTP induction.

XBD173 applied at the same concentration that exerts protection against H/H excitotoxicity (300 nM) did not affect LTP (143 ± 11% vs. control 149 ± 10%, *n* = 8/8, *p* = 0.148) in WT animals (Figure 4A,B). Moreover, when applied to slices of δ-KO (147 ± 21% vs. control 140 ± 8%, *n* = 9/9, *p* = 0.496), the potentiation was not significantly modulated (Figure 4C). Hence, XBD173 300 nM does not produce detrimental effects on LTP in these genotypes.

As THDOC and allopregnanolone are potential biosynthesised candidates released upon XBD173-induced TSPO activation, we further evaluated the effects of these neurosteroids on LTP. At a concentration of 100 nM, THDOC did not alter the potentiation, either in WT mice (146 ± 24% vs. control 145 ± 9%, *n* = 6/6, *p* = 0.562) (Figure 5A,B) or in δ-KO animals (144 ± 11% vs. control 142 ± 11%, *n* = 9/8, *p* = 0.820) (Figure 5C). Furthermore, allopregnanolone did not modulate LTP when applied at 300 nM to WT slices (162 ± 37% vs. control 158 ± 16%, *n* = 10/6, *p* = 0.695) (Figure 5D,E) or in δ-KO animals (154 ± 18% vs. control 140 ± 9%, *n* = 9/9, *p* = 0.055) (Figure 5F).

### 2.3. Recoding of Spontaneous Inhibitory Postsynaptic Currents

Finally, we examined the effectiveness of neurosteroids as positive allosteric modulators on pharmacologically isolated GABA_A_-receptor-mediated record spontaneous inhibitory postsynaptic currents (sIPSCs) in CA1 neurons by employing whole-cell patch-clamp recordings. Different synaptic currents parameters such as decay time and amplitude were analysed. 

XBD173 application at 300 nM resulted in an increase in synaptic transmission; specifically, sIPSC’s decay time (left; 30 ± 2% vs. control 29 ± 2%, *n* = 10/6, *p* = 0.037) and amplitude (right; 50 ± 16% vs. control 39 ± 10%, *n* = 10/6, *p* = 0.049) were significantly augmented (Figure 6A). After administration of THDOC at 100 nM, both decay time (left; 34 ± 2% vs. control 31 ± 4%, *n* = 12/8, *p* = 0.001) and amplitude of sIPSC (right; 54 ± 12% vs. control 39 ± 10%, *n* = 12/8, *p* = 0.002) were significantly increased (Figure 6B). Exposure to allopregnanolone at 100 nM resulted in a significant prolonged sIPSC decay time (left; 34 ± 1% vs. control 30 ± 2%, *n* = 8/6, *p* = 0.008), whereas amplitude was not affected (right; 40 ± 17% vs. control 35 ± 11%, *n* = 8/6, *p* = 0.250) (Figure 6C).

## 3. Discussion

The present study revealed that XBD173-induced neurosteroidogenesis and application of the potentially synthesised neurosteroids THDOC and allopregnanolone do not depress LTP at concentrations conferring neuroprotection in a model of ischaemia-induced excitotoxicity. Employing genetically modified mice, we discovered that neuroprotection triggered by XBD173 depends on TSPO and on δ-containing GABA_A_ receptors. Furthermore, we described that α5-GABA_A_ receptor subunits are a major molecular target of midazolam-induced neuroprotection.

Previous studies revealed that neurosteroids bind with high sensitivity to the δ-containing GABA_A_ receptors, and the hypnotic and anxiolytic effects of neurosteroids were substantially reduced when δ-KO mice were tested [27]. Specifically, the δ-subunit is directly related to the effects caused by THDOC administration, since this GABA_A_ receptor subtype was found to be responsible for the enhancement of sIPSC decay time in the presence of THDOC measured in cerebellar granule cells [28] and for the potentiation of acute tonic conductance, which is linked to neuroprotection and processes regarding hippocampal-related cognition [29]. Furthermore, it is known that both phasic (synaptic) and tonic (extrasynaptic) currents are sensitive to neurosteroid modulation [30]. Synaptic transmission modulation after neurosteroid application can be examined via sIPSC monitoring. Consistent with this, the present study demonstrates that neurosteroids released upon XBD173 application and direct administration of THDOC and allopregnanolone elevate GABA_A_-receptor-mediated activity. XBD173 and THDOC increased both decay time and amplitude, whereas allopregnanolone only prolonged the sIPSC decay time. Since these results are consistent with already published investigations [31], we may conclude that neurosteroids are able to modulate isolated GABAergic synaptic transmission.

Since recordings were made at room temperature (RT), we expanded the ischaemic period to 25 min. Neuroprotection was examined by exposing slices to oxygen–glucose deprivation, mimicking the in vivo situation when glucose and oxygen supplies are reduced or even eliminated. Early electrophysiological investigations in rat hippocampal slices already described a decline in synaptic activity and CA1 synaptic transmission after a period without oxygen and glucose [32]. A recent paper revealed an improvement in neuronal survival upon XBD173 treatment after a transient ischaemia in retinal cells [33], implying beneficial effects after an ischaemic event. In the present study, administration of XBD173 was unable to reverse the ischaemic-induced suppression of fEPSP slopes in δ-KO and TSPO-KO mice, suggesting that XBD173 induced neurosteroidogenesis via TSPO-induced release of neurosteroids, which target δ-containing GABA_A_ receptors. In accordance with our results, a recent study found that modulation of δ-containing GABA_A_ receptors exhibited neuroprotective effects after stroke and inflammation in mice [34]. Moreover, several reports described that extrasynaptic receptors (δ- or α5-containing GABA_A_ receptors) provide tonic conductance in the presence of low concentrations of ambient GABA [35], and this inhibition has been linked to neuroprotection [29].

To better understand the mechanism through which XBD173 exerts neuroprotection, it is important to identify the neurosteroids that are biosynthesised after TSPO activation. Unfortunately, a detailed analysis of the neurosteroids released upon its activation is still missing. THDOC and allopregnanolone have been reported to be synthesised after XBD173 application [24], acting as positive allosteric modulators at the GABA_A_ receptor with a similar potency and efficacy for the different receptor subtypes, thereby modulating a broad range of actions in the central nervous system [36]. Moreover, the two neurosteroids show higher potency when GABA_A_ receptor complexes contain the δ subunit [37]. Here, we show that at 100 nM, THDOC and allopregnanolone inhibited H/H-induced fEPSP slope suppression in WT mice. Our results indicate that THDOC and allopregnanolone may act as neuroprotective agents, most likely via enhancing δ-containing GABA_A_ receptors. 

XBD173 produces anxiolysis via neurosteroid synthesis and subsequent GABA_A_ receptor potentiation. In the present study, we show that administration of XBD173, THDOC, or allopregnanolone did not affect LTP (see Izumi et al. [38]). These results suggest that XBD173-induced neurosteroidogenesis and the putatively released neurosteroids THDOC and allopregnanolone do not inhibit physiological processes related to learning and memory. In contrast, we previously showed that a low nanomolar concentration of midazolam blocked LTP, mainly via α1-GABA_A_ receptor subunits [13]. At the same concentration, midazolam is effective as a sedative and anxiolytic, bearing the risk of producing cognitive disturbances in patients. Thus, TSPO may represent a promising target for the development of fast-acting anxiolytics without the side-effects inherent to benzodiazepines, e.g., sedation, tolerance development, abuse liability, and interference with cognitive performance. Additionally, a Phase II study has been conducted by Novartis in patients with generalised anxiety disorder (ClinicalTrials.gov identifier: NCT00108836) demonstrating the efficacy, safety, and tolerability of XBD173.

Neurosteroids and TSPO ligands are effective in models of traumatic brain injury, epilepsy, and stroke [21,39]. In contrast to benzodiazepines where utility in the chronic treatment of epilepsy is limited by tolerance, anticonvulsant tolerance is not observed with neurosteroids [40]. Thus, neurosteroids have the potential to be used in the chronic treatment of epilepsy. In particular, TSPO ligands might be useful for preventing secondary pathophysiological consequences and neuronal loss after traumatic, excitotoxic, or ischaemic brain damage. The development of cognitive deficits has not been reported for TSPO ligands and neurosteroids so far. Because of their clinical potential [41], neurosteroids and protein ligands continue to attract attention for the treatment of neurological as well as affective disorders [24].

Altogether, our findings suggest that XBD173 activates TSPO, thereby promoting the release of THDOC and allopregnanolone, and, via enhancing activity of δ-containing GABA_A_ receptors, mediate neuroprotection without affecting LTP. Even though midazolam is a very potent neuroprotective agent, it interferes with LTP [13] and this may provide an explanation for its detrimental effect on cognition. XBD173 may represent a promising alternative in perioperative anaesthesia with a less severe side-effect profile than that of benzodiazepines.

## 4. Materials and Methods

### 4.1. Animals

All procedures were performed in accordance with the German law on animal experimentation and were approved by the animal care committee (Technical University Munich, Munich, Germany). 

Mice (6–10 weeks old) from both sexes were used. The WT (C57BL/6) mice were obtained from Charles River (Italy) and the knock-in (KI) mouse lines α5-KI and α1/2/3-KI from Calco (Italy). The transgenic lines harbour a histidine (H) to arginine (R) point mutation in the benzodiazepine binding site, which renders the modified receptors insensitive to positive allosteric modulation by benzodiazepines. Importantly, responses to GABA remain completely unaltered [42]. 

The GABAδ knock-out (δ-KO) mouse line was bred by our group in Munich (Germany). This transgenic line was selected because it displays a decrease in the sensitivity to the neurosteroids’ sedative and anxiolytic effects [43], implying that the endogenous neurosteroidogenesis is not possible in this genotype.

### 4.2. Acute Brain Slice Preparation

Before decapitation, mice were deeply anaesthetised with isoflurane. The mouse brain was rapidly removed from the head and placed in ice-cold Ringer solution, and the brain was cut into sagittal hippocampal slices (350 µm thickness) at 4 °C. Slices were recovered for 30 min at 34 °C in a chamber submerged with artificial cerebrospinal fluid (aCSF) and bubbled with carbogen. Afterwards, the slices recovered for 1 h at RT (21–23 °C) before being transferred to the recording chamber (for details, see Puig-Bosch et al. [13]). All experiments were performed at RT.

### 4.3. Oxygen–Glucose Deprivation Model: H/H Measurements

We used an oxygen–glucose deprivation technique that mimics the conditions of an ex vivo ischaemic stroke [44] to study the underlying molecular mechanisms for neuroprotection improvement after H/H-induced excitotoxicity. For monitoring of the oxygen–glucose deprivation effects on hippocampal slices, i.e., to quantify neuronal damage, we established a protocol to record fEPSP slopes after an ischaemic period. The slope of fEPSP was recorded in the hippocampal CA1 *stratum radiatum*, induced by stimulation in the Schaffer collateral-associational commissural pathway of the same region and by using glass micropipettes filled with aCSF. fEPSP was evoked via an alternating test stimulus (50 µs, 5–20 V) using a bipolar tungsten electrode placed on the side of the recording pipette. A stable baseline was recorded and the last 20 min before the 25 min of ischaemic induction were averaged (compounds were washed-in for 1 h). At this point, the brain slices were perfused with a glucose-free aCSF (glucose was substituted for an equimolar concentration of D-mannitol) and deoxygenated with 95% N_2_/5% CO_2_. These changes were necessary to eliminate all glucose and oxygen remnants. After these 25 min, normoxic conditions were restored, aCSF was substituted with the one previously used, and the monitoring of the fEPSP slope continued for 60 min to measure the recovery. fEPSP slopes from the last 10 min of the 1 h recovery period were averaged and normalised to the 20 min baseline before H/H. Different protocols to induce ischaemic conditions exist and they may depend, for instance, on the temperature at which they are performed. Since our experiments were executed at RT, physiological processes occur slower than at 37 °C and the time of 25 min chosen for the experiments mirrors this phenomenon. Moreover, oxygen and aCSF retrieval does not immediately affect the brain slices. Thus, the hippocampus only deteriorates as time passes in the presence of glucose-free fluid and N_2_, but it does not directly die.

### 4.4. Long-Term Potentiation Recordings

The slope of fEPSP was recorded in the hippocampal CA1 *stratum radiatum*. For the slope formation, we followed the H/H measurement protocol explained above, but using two stimulating electrodes that were placed on both sides of the recording pipette. With this electrode composition, we could stimulate the non-overlapping populations of fibres of the Schaffer collateral-associational commissural pathway. After recording a stable baseline (fEPSP slope of around 25–30% of the maximum response) with the WinLTP program, LTP was induced by delivering a high-frequency stimulation train (100 pulses at 100 Hz during 1 s) through one of the two stimulating electrodes (for details see Puig-Bosch et al. [13]). After the delivery of the stimulation without any substance, the potentiation of the fEPSP slope was recorded for 60 min after the tetanic stimulus, conserving the settings used for the baseline. Afterwards, XBD173, THDOC, or allopregnanolone were applied in the flowing solution for 60 min before inducing LTP in the second input following high-frequency stimulation, which was delivered through the second electrode. The fEPSP slope was calculated between 20–80% of the peak amplitude and then normalised to the 20 min baseline before stimulation. LTP inhibition or blockage was defined when the fEPSP slope in the last 10 min of the 1 h recovery period after high-frequency stimulation was 20% lower than the pre-stimulation slope.

### 4.5. Whole-Cell Patch-Clamp Recordings 

Whole-cell patch-clamp recordings were also obtained from CA1 hippocampal pyramidal cell neurons as previously described [13].

The recording electrode was positioned on a CA1 pyramidal neuron with a micromanipulator, and cells were held at −70 mV to sIPSC. To isolate GABA_A_-mediated currents, the following antagonists were used: D-AP5 (50 μM) and NBQX (5 μM) to respectively block NMDA and AMPA receptors and CGP55845 (5 μM) to block GABAB receptors. 

To monitor sIPSC, control recordings were performed for 6 min after ensuring that the cell was healthy and recorded again after 30 min of washing with XBD173, THDOC, or allopregnanolone. At the end of the measurement, the GABA_A_ receptor antagonist bicuculline (10 μM) was applied and the sIPSC events were eliminated, verifying that the measurements were pharmacologically isolated GABA_A_-receptor-mediated currents.

### 4.6. Experimental Design and Statistical Analysis

No explicit randomisation or blinding methods were used to assign animals to experimental conditions. The *n* value is shown as x slices out of y animals, e.g., *n* = 9/7, where the first is the number of slices and the second the number of animals used for a certain experiment. The sample size was determined based on previous experience and at most two slices per animal were used, by assuming that slices were independent within animals. Statistical analysis and graphical design were conducted with GraphPad Prism 6.01 (GraphPad Software, USA). Given the small sample size, it was not possible to check for data normality; hence the corresponding non-parametric tests were applied. For comparisons between two groups, the paired Wilcoxon test was used to statistically analyse LTP, and patch-clamp experiments (linked samples) and the unpaired Mann–Whitney test were employed for H/H recordings because control and drug testing were undertaken in two different brain slices (unlinked samples). All data from LTP and patch-clamp experiments, and time course data from ischaemic experiments, are shown as mean ± SD%. However, H/H experimental data from the last 10 min of the 1 h recovery period are reported as median (IQR)%, where IQR is the interquartile range subtracted from the difference between the third and first quartile (IQR = Q3−Q1). This data reporting is more precise in this specific case because these values are non-normally distributed and independent samples.

Differences were considered statistically significant when *p* < 0.05, and they are indicated with an asterisk (*) in the figures.

## Figures and Tables

**Figure 1 ijms-24-09056-f001:**
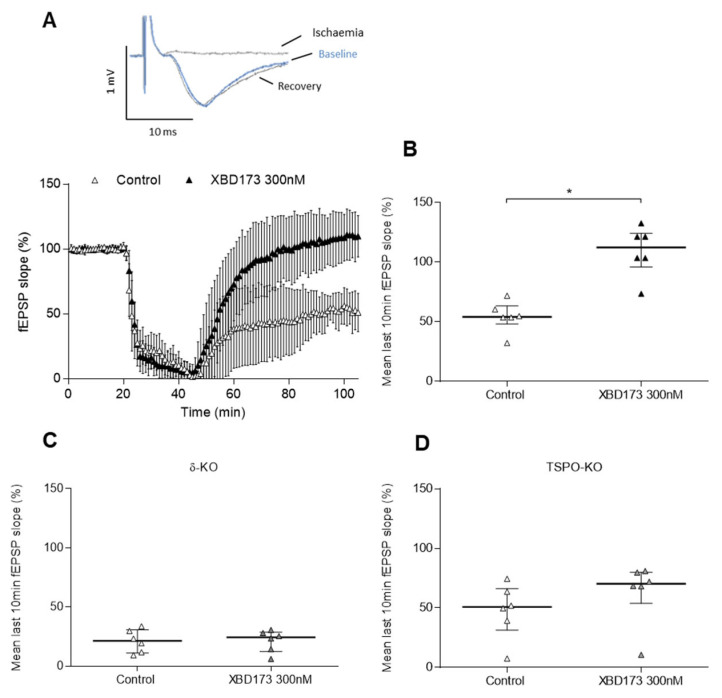
Effect of XBD173 on ischaemic experiments in WT, δ-KO, and TSPO-KO mice. (**A**) The insets show representative fEPSP slope traces during the baseline, the ischaemia, and the recovery periods in the presence of 300 nM XBD173. Each symbol represents the averaged fEPSP slopes (mean ± SD) and all responses were normalised to the baseline recorded for 20 min prior to the ischaemic period started for control and XBD173 300 nM in the WT genotype. (**B**–**D**) Scatter plots summarising the last 10 min of the 1 h recovery period after ischaemia, where median (IQR)% are represented. XBD173 at 300 nM significantly increased the fEPSP slope from the recovery in (**B**) the WT genotype (112(28)% vs. control 54(15)%, *n* = 6/6, *p* = 0.002). In contrast, (**C**) in the δ-KO genotype, the fEPSP slope from the recovery went back to baseline levels (24(16)% vs. control 21(19)%, *n* = 6/6, *p* = 0.132), as well as in the (**D**) TSPO-KO genotype (70(35)% vs. control 51(23)%, *n* = 6/6, *p* = 0.898). Statistically significant differences are marked with an * for *p* < 0.05 when the unpaired Mann–Whitney test was applied.

**Figure 2 ijms-24-09056-f002:**
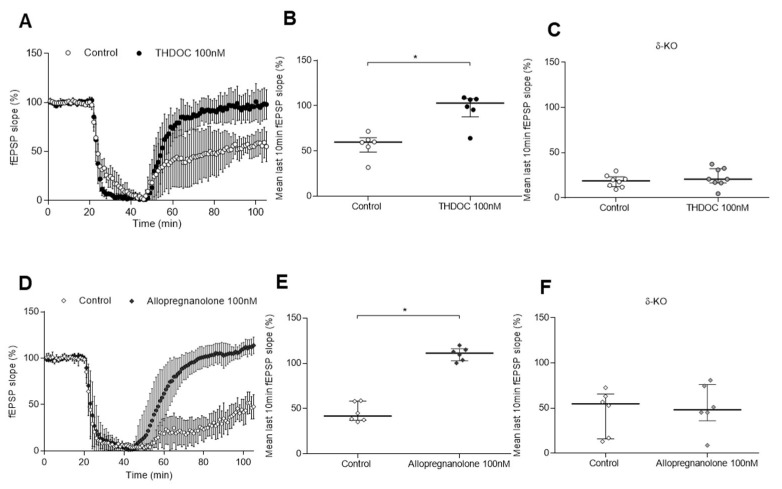
Effect of THDOC and allopregnanolone on ischaemia experiments in WT and δ-KO mice. (**A,D**) Each symbol represents the averaged fEPSP slopes (mean ± SD) and all responses were normalised to the baseline recorded for 20 min prior to the H/H period started for control and (**A**) THDOC and (**D**) allopregnanolone 100 nM in the WT genotype. (**B**,**C**,**E**,**F**) Scatter plots summarising the last 10 min of the 1 h recovery period after ischaemia, where median (IQR)% are represented. (**B**) In WT, THDOC 100 nM (103(20) % vs. control 60(16) %, *n* = 6/6, *p* = 0.004) significantly increased the fEPSP slope during recovery. (**C**) In δ-KO, the fEPSP slope after THDOC 100 nM exposure (21(16) % vs. control 19(11) %, *n* = 8/8, *p* = 0.375) went back to baseline levels. (**E**) In WT, allopregnanolone 100 nM (111(15) % vs. control 42(22)%, *n* = 6/6, *p* = 0.002) significantly increased the fEPSP slope during the recovery, (**F**) whereas in δ-KO, no recovery was seen after allopregnanolone administration (48(40) % vs. control 55(49)%, *n* = 6/6, *p* > 0.999). Statistically significant differences are marked with an * for *p* < 0.05 when the unpaired Mann–Whitney test was applied.

**Figure 3 ijms-24-09056-f003:**
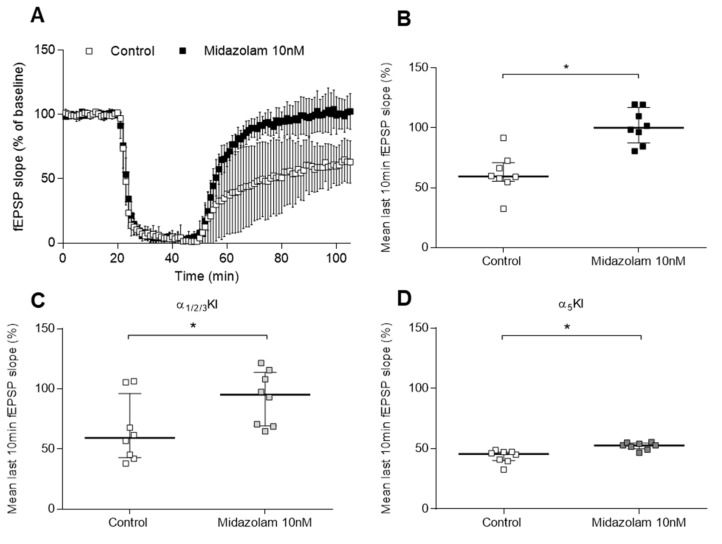
Effect of midazolam on ischaemia experiments in WT, α1/2/3-KI, and α5-KI mice. (**A**) Each symbol represents the averaged fEPSP slopes (mean ± SD) and all responses were normalised to the baseline recorded for 20 min prior to the H/H period started for control and midazolam 10 nM in WT genotype. (**B**–**D**) Scatter plots summarising the last 10 min of the 1 h recovery period after ischaemia, where median (IQR)% are represented. Midazolam at 10 nM significantly augmented the fEPSP slope from the recovery in (**B**) WT (100(29) % vs. control 59 (15) %, *n* = 8/7, *p* = 0.001) and (**C**) α1/2/3-KI genotype (95(44) % vs. control 59(53) %, *n* = 8/8, *p* = 0.028). (**D**) In the α5-KI genotype, the fEPSP slope from the recovery went back to baseline levels (53(5) % vs. control 46(7) %, *n* = 8/7, *p* = 0.001); however, a significant increase was apparent. Statistically significant differences are marked with an * for *p* < 0.05 when the unpaired Mann–Whitney test was applied.

**Figure 4 ijms-24-09056-f004:**
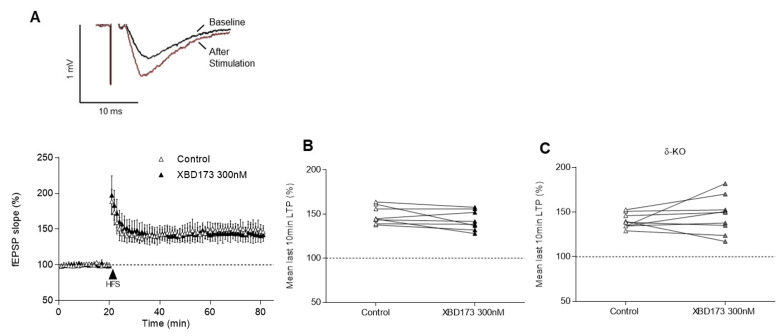
Effect of XBD173 after LTP induction in WT and δ-KO mice. (**A**) The insets show representative fEPSP slope traces before and after high-frequency stimulation in the presence of 300 nM XBD173. Each symbol represents the averaged fEPSP slopes (mean ± SD) and all responses were normalised to the baseline recorded for 20 min prior to delivering the tetanic stimulus (arrow) before and after 60 min of XBD173 300 nM. (**B**,**C**) Connected scatter plots of the fEPSP slope from min 50 to 60 after high-frequency stimulation. XBD173 did not significantly alter LTP in (**B**) WT (143 ± 11% vs. control 149 ± 10%, *n* = 8/8, *p* = 0.148) or in (**C**) δ-KO (147 ± 21% vs. control 140 ± 8%, *n* = 9/9, *p* = 0.496) genotypes. No significant differences were observed when the Wilcoxon test was applied.

**Figure 5 ijms-24-09056-f005:**
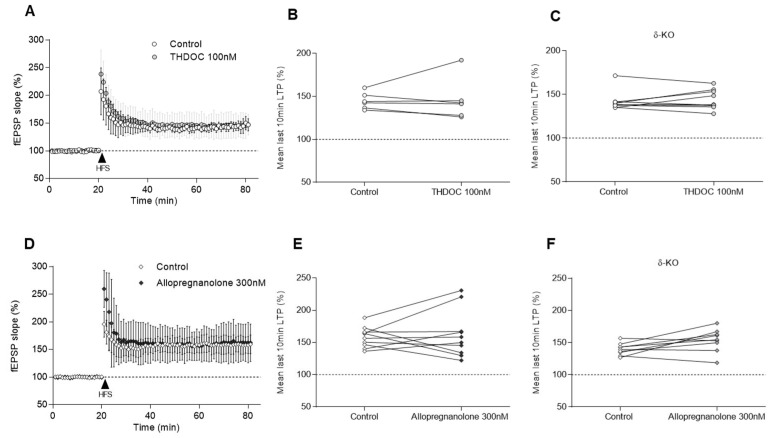
Effect of THDOC and allopregnanolone after LTP induction in WT and δ-KO mice. (**A**,**D**) Each symbol represents the averaged fEPSP slopes (mean ± SD) and all responses were normalised to the baseline recorded for 20 min prior to delivering the tetanic stimulus (arrow) before and after 60 min of (**A**) THDOC 100 nM and (**D**) allopregnanolone 100 nM exposure. (**B**,**C**,**E**,**F**) Connected scatter plots of the fEPSP slope from min 50 to 60 after high-frequency stimulation. THDOC 100 nM did not modify LTP either in (**B**) WT mice (146 ± 24% vs. control 145 ± 9%, *n* = 6/6, *p* = 0.562) or in (**C**) δ-KO animals (144 ± 11% vs. control 142 ± 11%, *n* = 9/8, *p* = 0.820). Moreover, after allopregnanolone administration in WT animals (162 ± 37% vs. control 158 ± 16%, *n* = 10/6, *p* = 0.695) and in (**F**) δ-KO mice (154 ± 18% vs. control 140 ± 9%, *n* = 9/9, *p* = 0.055), no significant LTP changes were evident. When the Wilcoxon test was applied, no significant differences were observed.

**Figure 6 ijms-24-09056-f006:**
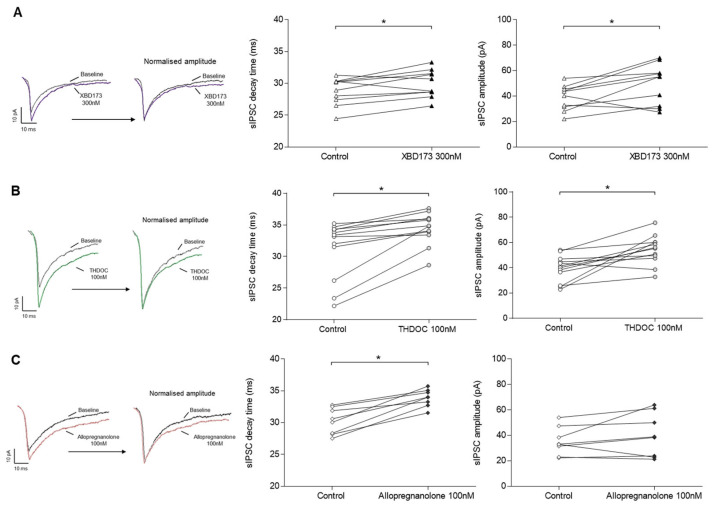
Effect of XBD173, THDOC, and allopregnanolone on sIPSCs. Whole-cell patch-clamp recordings in WT mice. Connected scatter plots of the sIPSC properties under control and post-drug conditions. Representative sIPSC traces for control and drug conditions, showing original overlapped traces (left) and overlapped traces with a normalised amplitude (right), are included for (**A**) XBD173, (**B**) THDOC, and (**C**) allopregnanolone. (**A**) After 30 min of XBD173 application at 300 nM, the sIPSC decay time was significantly increased (left; 30 ± 2% vs. control 29 ± 2%, *n* = 10/6, *p =* 0.037), as well as the amplitude (right; 50 ± 16% vs. control 39 ± 10%, *n* = 10/6, *p =* 0.049). (**B**) After 30 min of THDOC exposure at 100 nM, the sIPSC decay time (left; 34 ± 2% vs. control 31 ± 4%, *n* = 12/8, *p* = 0.001) and the amplitude (right; 54 ± 12% vs. control 39 ± 10%, *n* = 12/8, *p =* 0.002) were significantly augmented. (**C**) After 30 min of allopregnanolone perfusion at 100 nM, the sIPSC decay time was significantly increased (left; 34 ± 1% vs. control 30 ± 2%, *n* = 8/6, *p =* 0.008) but not the amplitude (right; 40 ± 17% vs. control 35 ± 11%, *n* = 8/6, *p =* 0.250). Statistically significant differences are marked with an * for *p* < 0.05 when the Wilcoxon test was applied.

## Data Availability

Data supporting the findings of this study are available from the corresponding author upon reasonable request.

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
