# Peer review of "Neurosteroids Mediate Neuroprotection in an In Vitro Model of Hypoxic/Hypoglycaemic Excitotoxicity via δ-GABAA Receptors without Affecting Synaptic Plasticity"

_ijms, 2023, doi:10.3390/ijms24109056_

Round 1

Reviewer 1 Report

This was a well-conceived and nicely presented study showing the protective effects of neurosteroids on recovery from hypoglycemia and hypoxia using the hippocampal brain slice model system.  Experiments appear to be well done and results support the Authors conclusions.  There were a few mistakes that will need correction, but otherwise this is a good paper.

“for causing advert side effects on cognition”

should be “for causing adverse side effects on cognition”

“2.3. Whole-cell patch-clamp recordings” should be removed, since no patch-clamp data is shown.

“Several studies currently investigate how this det-rimental effect could overcome.”  This sentence should be removed since it makes no sense and is not needed.

“4.5. Long-term potentiation recordings

Whole-cell patch-clamp recordings were also obtained from CA1 hippocampal pyramidal cell neurons as previously described [13].”  This should be removed unless the Authors decide to add patch-clamp results to their paper.

Some improvements could be made.

Author Response

Thank you for the feedback. We'll address the comments from Reviewer 1 with a point-by-point response:

1. “for causing advert side effects on cognition” should be “for causing adverse side effects on cognition

Response: We modified the sentence according to the reviewer’s 1 suggestion.

2. “2.3. Whole-cell patch-clamp recordings” should be removed, since no patch-clamp data is shown.

Response: The title of section 2.3 has been modified to “Recording of spontaneous inhibitory postsynaptic currents” since it may be more appropriate. However, the entire date described in chapter 2.3. and in Fig.6 are single-cell patch-clamp experiments conducted in whole-cell mode. We also changed the caption of Fig.6 : “Figure 6. Effect of XBD173, THDOC and allopregnanolone on sIPSCs. Whole-cell patch-clamp recordings in WT mice. Connected scatter….”

We noticed that the chapter 4.5. in the method section, which should describe whole-cell patch-clamp recordings, bears the wrong heading. We exchanged “Long-term potentiation recordings” for “Whole-cell patch-clamp recordings.

3. “Several studies currently investigate how this detrimental effect could overcome.” This sentence should be removed since it makes no sense and is not needed.

Response: This sentence has been removed.

4.  “4.5. Long-term potentiation recordings

Whole-cell patch-clamp recordings were also obtained from CA1 hippocampal pyramidal cell neurons as previously described [13].”  This should be removed unless the Authors decide to add patch-clamp results to their paper.

Response: We apologize for this mistake and are grateful for the referee’s comment. The title of section 4.5 has been changed to “Whole-cell patch-clamp recordings” since the previous title was completely wrong. As we mentioned above, there are patch-clamp experiments displayed in section 2.3 and Fig. 6. Thus, in our opinion, it is appropriate to keep this modified text:

Whole-cell patch-clamp recordings

Whole-cell patch-clamp recordings were also obtained from CA1 hippocampal pyramidal cell neurons as previously described [13]”.

Comments on the Quality of English Language

Some improvements could be made.

Response: To improve the quality of the English language, we have used an online English proofreading service to make some refinements on the general writing of the manuscript. Furthermore, we have read the manuscript several times to detect typos.

Reviewer 2 Report

Here I present my comments on the work entitled Neurosteroids mediate neuroprotection in an in vitro model of hypoxic/hypoglycaemic excitotoxicity via delta-GABAA receptors without affecting synaptic plasticityby Puig-Bosch et al. for publication in the International Journal of Molecular Sciences (Manuscript ID: ijms-2385525).

The manuscript is very well written and it is presented by the authors in a very comprehensive manner. Experiments and interpretations are sound and the figures and data are displayed clearly. From what I see, statistics are sound as well as animal cohorts and experimental repetitions are correctly chosen. Further evaluations on the overall significance of this study could be further judged by the general reader and be studied by the authors in the future.

 My final opinion is that, based on the technical quality, style, and scientific rigor of this manuscript, it could be accepted for publication in its current state.

Author Response

We appreciate the feedback received from Reviewer 2 and we’ll consider the mentioned comments for upcoming studies.